# Reality and Future of Interculturality in Today's Schools

David Pérez-Jorge *, Ana Isabel González-Herrera, Miriam González-Afonso and Anthea Gara Santos-Álvarez

Department of Didactics, Educational Research University of La Laguna, 38200 Tenerife, Spain; agonzale@ull.edu.es (A.I.G.-H.); mcglez@ull.es (M.G.-A.); asantosa@ull.edu.es (A.G.S.-Á.)
* Correspondence: dpjorge@ull.edu.es

**Abstract:** In today's society, high-quality educational contexts must include intercultural education and educational inclusion as main elements of school culture. Equity, social justice, and equal opportunities for everybody require the construction of flexible processes, relationships, and organizational structures open to diversity. This paper presents an updated review of studies focusing on models of educational responses adapted to cultural diversity. It analyzes the response of schools as collaborative communities in intercultural education and their reality as inclusive and intercultural communities. An exhaustive search of documents was carried out, consulting the Web of Science (WoS), Scopus, and Dialnet databases. After analyzing and applying inclusion and exclusion criteria, 21 documents were identified that showed the structural, cultural, and relational transformation of educational centers and the improvement of their organizational and public response and adaptation to current needs. The challenge of building intercultural educational contexts is a concern for schools and the educational agents who coexist in them.

**Keywords:** intercultural education; interculturality; school; school communities; diversity





## 1. Introduction

Since 1990, interculturality has been an important concept worldwide. It has progressively gained prominence in the public, social, and political spheres as the focus of interest of educational improvements at both national institutional and inter/transnational levels [1,2]. If we focus on the educational field, schools have become an important part of intercultural action [3].

Interculturality is increasingly used as a strategy to promote reflection and a culture of dialogue in educational institutions. There is a clear orientation towards changing certain hegemonies of legitimization of the knowledge of what is taught and how it is taught in schools. The aim is to transform those organizational and structuring routines of the school institution (school organization and classroom grouping) that reproduce exclusion. This circumstance, which can be linked to the historical struggles of social movements and their demands for social justice, "can also be seen, at the same time, from another perspective: the one that links it to the global designs of power, capital and market" [1] (p. 2). It is therefore necessary to understand the current complex situation surrounding the concept of interculturality in formal education. In this sense, given that the term "interculturality" seems to have been used in multicultural, neoliberal discourses, policies, and strategies in an assimilationist and bureaucratic tone, it is useful to understand that "functional interculturality" (derived from neoliberal logic) unquestioningly complies with the established hegemonic rules. On the other hand, "critical interculturality" as a political, social, epistemic, and ethical project suppresses the causes of social asymmetry and cultural discrimination that hinder the possibilities of authentic cultural inclusion.

In the field of education, the influence of the neoliberal discourse has tinged the battles for social justice that emerged in the pedagogies of the oppressed [4] and, later, of hope [5,6]. Those that [7,8] were later raised as banners concerning the transformation of

the established system refs. [1,2] highlighted the usefulness of most educational reforms and pointed out that these reforms, coinciding with neoliberation policies at different historical moments and in contexts close to globalization which recognize the multiethnic and multilingual character of countries, introduced specific policies for "the different" as part of this multicultural logic of transnational capitalism.

### 1.1. The Critical Approach to Interculturality in Different Contexts

Based on this socio-critical historical perspective, it is useful to point out the divergences, different nuances, and particularities in the way in which each context has developed policies and instruments of public action to respond to the intercultural phenomenon [9]. Conceptions of interculturality differ substantially depending on the countries on which the analysis is focused on. As it was pointed out in [10], the notion of interculturality and intercultural education is not the same in countries that have been created thanks to the migration of diverse cultures, such as Canada, as it is in other European countries and the United States, which have migrations from other countries with a lower socioeconomic level or status. The author also discussed other countries, such as Mexico, where multiculturalism is part of their history and where "cultural differences between the dominant group and the indigenous cultures have led to everything from extermination attempts to concerted national assimilation efforts" [10] (p. 11). In analyzing intercultural education in terms of the "paradoxical closeness" between the European and Latin American contexts, ref. [11] argued that in both territories coexist "different models and approaches to tackle cultural diversity, which, far from contributing to the transformation of schools and society, tend to deepen social inequalities" [11] (p. 38). This has happened in many countries because the effectiveness of intercultural proposals depends not only on how much the actions generated are consistent with the needs of each educational context but also on other structural measures that transcend the school environment. If these conditions are not complied with, the "so-called intercultural initiatives could be used as a camouflage for inequalities, as an alibi to avoid the adoption of initiatives that are genuinely respectful of cultural diversity, or as a mere showcase for the most "touristy" and superficial vision of cultural manifestations" [12] (p. 45).

In this respect, it is believed that the European Union's policy initiatives have focused less on the positive effect of improved schooling, or on the political and social participation of immigrant students, but rather on cohesion. The logic governing European policy documents is based on achieving better integration of immigrants into society and, more specifically, into the labor market [13]. In this sense, the authors pointed out that, at these governmental levels, the empowerment of the social and political participation of this group through a more inclusive education is only addressed as a "side effect" of the policies developed. This is consistent with [14] analysis of the clear inconsistencies between the statements of European rhetoric, which stress democracy and human rights, and what is emphasized by national educational policies, basically focused on the acquisition of language and national values to "fit in" migrants. These policies, which supposedly promote social justice, may fail to achieve inclusive citizenship if, while acknowledging demographic changes, they neglect everyday injustices and the European historical imprint of racism and xenophobia [15,16].

Until recently, it would seem that what has prevailed is to find channels, different in each country, to integrate the "different" from the educational system and making them adapt to the "majority". However, there are other indications that their potential can contribute very much to the development of intercultural competence of all citizens due to their capacity for resilience, acceptance, and openness to novelty, especially in second- and third-generation migrant citizens [17]. As [18] stated, the potential of the individual to demonstrate intercultural competence increases as the experience of cultural diversity becomes more sophisticated, moving from ethno-centrism to ethno-relativism. It is in this sense that situations of cultural dialogue can be fostered, based on the empathy and flexibility that are associated, in many cases, with the development of intercultural

competence. As [19] defines intercultural competence as "the ability to interact effectively and appropriately in intercultural situations, based on one's intercultural knowledge, skills and attitudes" (p. 247), what leads to intercultural competence is intercultural learning. In this regard, Cortazzi and Jin (2013) stated that intercultural learning refers to how we understand other cultures and our own culture through interaction, generating a cultural synergy that "yields a holistic benefit that is larger than separated elements. It suggests the importance of peer dialogue as well as teacher-student discussion about ways of learning to develop local, contextualized ways of learning." [20] (p. 2).

*1.2. Evolution of the Concept of Intercultural Education*

All this conceptualization emerged in the field of analysis that has been forged around the central subject of this study, which is intercultural education. The expression "intercultural education" first appeared within the European context in 1983 when the European legislative framework emphasized the intercultural dimension of education in a resolution for the schooling of immigrant children [21]. Since then, as developed by [22], there has been a clear evolution in the conception and approach given to the expression. Since 2015, research on intercultural education begins to move towards a more dynamic, ethical, and transcultural perspective, which is brought to educational practice through interaction and cultural dialogue [23,24]. The static view of intercultural education as a skill in dealing with other cultures, ethnicities, or languages has been disregarded. It has become increasingly evident that cultural diversity itself, promoted by national education systems, is not appropriate for intercultural learning and dialogue to take place [25,26], always in a living process of formation. In this field, where discourses and training practice, sometimes unrelated, have multiplied worldwide, a rallying point is extracted: intercultural education, supported by inclusion and social justice, should address the entire school community, fostering dialogue and democratic coexistence, going beyond the mere recognition and acceptance of differences [27,28]. The authors of [22] highlight an important contradiction: the national education system and the teaching and learning approach it fosters is precisely what raises obstacles to intercultural dialogue. Thus, it is easy to understand that the school recognizes this intercultural approach in declarative terms but maintains an assimilationist logic in its praxis [29]. The conflict between the official discourse on intercultural education and its actual application in school practice should not be overlooked.

In specific areas, where social coexistence with diverse cultures has significantly intensified (as is the case in the Spanish context), publications have also increased significantly in the last two decades [30–32]. One of the fundamental reasons concerns the interest in caring for the intercultural coexistence that has developed given the high rate of immigration in Spain. According to [12], there is a clear definition of the concept of intercultural education that emphasizes the perspective of interaction and the benefit of cultural pluralism typical of democratic societies as a pedagogical resource. Intercultural pedagogy is understood as a "reflection on education, [ . . . ] based on the recognition of cultural diversity value" [12] (p. 45). What is important about this proposal is that it is based on a model of analysis and intervention that includes all the dimensions of the educational process. At the time when intercultural education was taking shape in Spain, a clear vision was provided showing the existing gaps in the research carried out up to that time in relation to both structural and curricular measures. The academic centers had a limited margin to put faith in an educational project based on a medium/long-term consolidated center where interculturality was hardly existent. At that time, it became clear that "organizational and curricular changes do not usually affect the general curriculum, but rather have a compensatory character" [12] (p. 46).

From the results of these studies, important recommendations were drawn regarding the grouping and selection of students. It was aimed at the need for flexible groups adapted to teaching–learning processes. In relation to formative and evaluative activities, cooperative learning was emphasized to favor interaction. Regarding another critical dimension, the participation of families and students in the center, it was suggested that

it is not so much a matter of continuing to facilitate family–school–community bridges, but rather of "creating educational spaces where all those involved have a role to play in the definition and development of the educational model" [12] (p. 49). Until 2003, intercultural education was an illusion rather than a reality; it was necessary to state and support actions towards the creation of models of educational projects in schools that would build a solid and consistent proposal for intercultural education. Later, in their systematic review, ref. [33] point out two fundamental lines that have marked a trend between 2009 and 2019: the one related to the perceptions and beliefs of teachers in relation to ethnocultural diversity, and on the other hand, the concern to offer a consistent and sustained response to this diversity in schools and communities [34,35], bearing in mind the crucial role of the family in the important dynamics of change both inside and outside the classroom.

*1.3. Towards an Inclusive Approach to Intercultural Education in Schools*

Since 2015, there has been a certain abundance of studies emphasizing an inclusive and dynamic conception of intercultural education based on quality interactions generated in schools. However, it is necessary to review what has happened in the last five years, analyzing whether this approach has permeated organizational and curricular processes and pedagogical practice in schools and whether the "terrain marked by political discourse" on the inclusion of cultural diversity [25] has developed. From the perspective that has been reflected, beyond neoliberal slogans, a review is proposed on the way educational institutions and school communities may be succeeding in inclusive practice and process practice towards creating an authentic intercultural education. This review considers that an inclusive intercultural approach must be supported by an involved and collaborative institutional organization, the progress of which is achieved through the construction of a sense of belonging to the educational community [36–40]. The need for collective responsibility in a climate of exchange between educational agents for the improvement of competent learning for all students [41] derives from this principle. The importance of transforming practice by reviewing the ways of understanding relationships and interactions between the staff, with the students, and with the educational community must be recognized [42]. From this perspective, it is important to assume a meso-, micro-, and macro- vision of intercultural education, analyzing whether the political purposes and the type of institutional conditions that educational administrations impose on schools are consistent with the daily margins of action of teachers in their centers. As [43] (p. 66) points out "the school, in the last decades, has been accumulating an excess of expectations and missions, as a consequence of having delegated everything that the family or society could not do or was concerned about". This takes the risk of making the school responsible for what it cannot achieve in isolation and on its own. In this sense, it is necessary to strengthen the conception of educational contexts as spaces for dialogue beyond the physical barriers of schools [44], where critical reflection on dilemmas and paradoxes can be enriched so that inclusion can take root in praxis [45]. It must be assumed that in order to achieve a true culture of diversity, the traditional view of cultures and the compensatory curricular approach for minorities must be overcome [25,44,46]. At the same time, intercultural education must be considered beyond specific ethnic markers, from which a process of homogenization of cultural groups is sought to be assembled starting from a cultural logic [4]. The message that [47] (p. 3) conveyed a few years ago in a monograph focused on intercultural education culminates the theoretical framing of this systematic review:

Today it is more necessary than ever to promote intercultural education with a pedagogical approach of inclusive nature and focused on the community, and it is not only a matter of addressing the organizational, didactic, and curricular responses of the intercultural school, but the nature, meaning, imprint and character of educational actions that generate peaceful and respectful coexistence, equity and participation, education and democratic values in academic institutions.

## 2. Objectives

Based on this approach, the aim of this systematic review is to examine the role of the school community in the implementation of authentic inclusive intercultural education.

The specific objectives are as follows:

(a) To conduct an updated review of the reorganization processes that the school community have determined in order to offer an inclusive educational response to cultural diversity.

(b) To analyze the current response of schools as participatory and collaborative communities in intercultural education.

(c) To evaluate the current practical reality of schools as inclusive and intercultural education centers.

## 3. Materials and Methods

To achieve the proposed objectives, a systematic literature review was carried out applying the protocol of the updated PRISMA statement [48] for the identification and selection of documents. This methodology aims to identify, analyze, and assess those studies that highlight the institutional dimension of schools as participatory and collaborative educational communities. The intercultural education approach is analyzed in relation to the following dimensions:

1. The construction processes of the participatory school climate.
2. The organizational and pedagogical structure of inclusive schools.
3. The impact of practice and experiences created in these inclusive intercultural schools to generate intercultural learning.

To study these facets, concepts related to intercultural education, cultural diversity, coexistence, and school climate, as well as the school context, were chosen as fundamental elements that make up a true educational community based on the model of collaboration and participation.

Scopus and Web of Science (WoS) databases were used for English language articles and Dialnet for Spanish ones. The following descriptors from the UNESCO thesaurus were used to search for documents: school, "intercultural education", participation, "cultural diversity", practice, "primary education", "secondary education", and "elementary education". These descriptors were combined with the Boolean operators AND, OR, and NOT. Several combinations were tried until the final search formula was: "intercultural education" AND (school OR "primary education" OR "secondary education" OR "elementary education"). The search was carried out using the topics in both English and Spanish.

After each search, the documents were selected based on the inclusion and exclusion criteria established beforehand (Table 1).

**Table 1.** Study inclusion and exclusion criteria.

| Inclusion Criteria | Exclusion Criteria |
| --- | --- |
| a. Documents published between 2018 and 2023. | a. Documents prior to 2018. |
| b. Documents in English or Spanish. | b. Documents in languages other than English or Spanish. |
| c. Documents in Open Access. | c. Studies that are not of a scientific nature. |
| d. Studies related to the educational, sociological, and psychological fields. | d. Highly specialized studies in religion, anthropology, language, or specific languages. |
| e. Studies focused on basic, primary, and secondary education in formal education in school contexts. | e. Studies in non-compulsory education and in other non-formal educational contexts. |
| f. Studies that apply a quantitative, qualitative, or mixed research method. | f. Analysis of documents, essays, or theoretical reflections. |
| g. Studies focused on intercultural education from the inclusive cultural diversity approach in schools. | g. Studies focused on other types of specific educational support needs unrelated to intercultural school contexts and with a compensatory and non-inclusive approach. |

For the analysis and classification of the documents based on the established dimensions (processes of construction of participatory school climate; organizational and pedagogical structure of inclusive schools and impact of the practice and experiences created in these inclusive intercultural schools to generate intercultural learning), an analysis was applied to evaluate the degree of agreement and disagreement among the 3 judges who participated in the classification of the documents based on the inclusion and exclusion criteria that were assessed. Considering that some agreements could be the result of chance and that their random effect could give a greater reliability effect than the real one [49], the Perreault and Leigh coincidence coefficient was used to amend this result. Its values range from 0 (no agreement) to 1 (perfect agreement), and the coincidence value was I = 0.83 which made it possible to consider the final selection of documents as adequate.

$$I_r = \left( \frac{Fo}{N} - \frac{1}{K} \right) \cdot \left( \frac{K}{K-1} \right)^{1/2} \text{ if } \frac{Fo}{N} \geq \frac{1}{K}$$

$$I_r = 0 \text{ if } \frac{Fo}{N} < \frac{1}{K}$$

*Fo* = number of opinions on which the judges agree
*N* = total number of opinions
*K* = total number of categories

*Procedure*

The exhaustive search began by using the keywords, then the inclusion criteria were applied, eliminating those documents that did not meet them. Using the Mendeley bibliographic manager, duplicates were located and subsequently eliminated from the initial selection. After this, titles and abstracts were read and the studies that met the inclusion and exclusion criteria were selected. A complete reading was performed out of the chosen texts, and those that did not focus on the objectives of the study after an in-depth analysis of their content were excluded. The objectives, sample, methodology, main results, and conclusions were analyzed for each article from the sample of selected documents.

The initial selection of documents was as follows:

A total of 3755 results were obtained. After applying the inclusion and exclusion criteria, 3732 documents were eliminated. The final sample of studies consisted of 21 documents. The following flow chart shows the selection process and the thorough examination of the articles (see Figure 1).

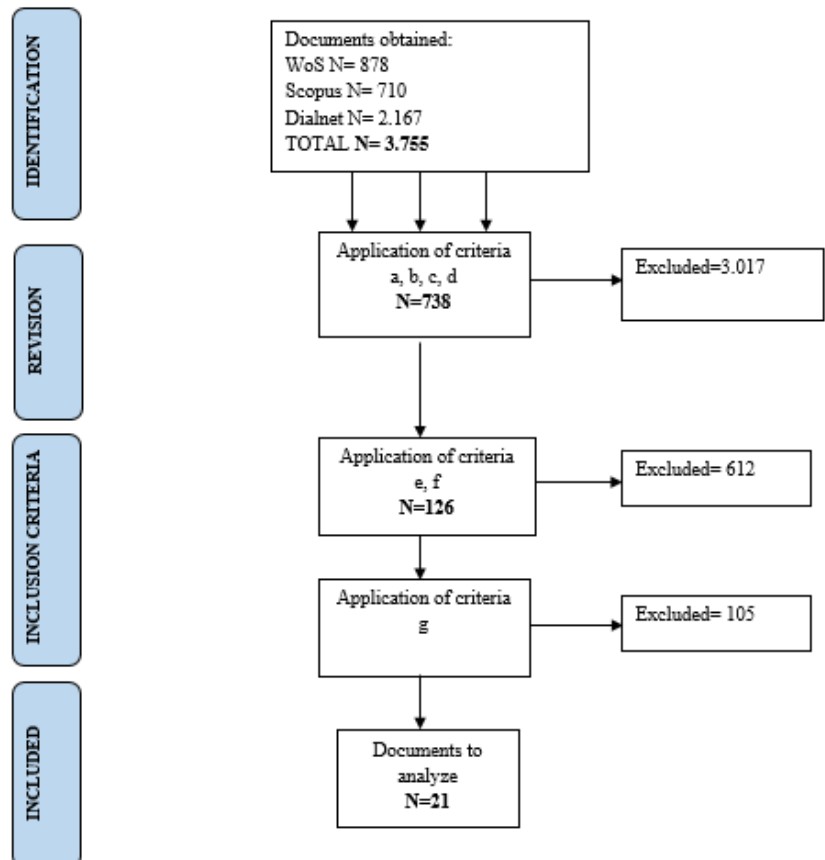

**Figure 1.** Flow chart of the selection process.

### 4. Results

*Characteristics of the Included Studies*

All articles published on the subject up to the time of the closing of this review (2 April 2023) were reviewed. The 21 articles selected for the review were studies published mainly in English between 2018 and 2023, which ensured results on experiences developed in educational centers around the new models of promotion and work of interculturality in schools. The studies were conducted mainly in Spain and Chile, although they were also carried out in countries such as Mexico, Brazil, Colombia, Russia, Italy, Austria, Denmark, Slovenia, Poland, and the United Kingdom). It is noteworthy that 11 out of the 21 studies were conducted exclusively in Spain.

Four studies used a purely quantitative methodology (19%; N = 4) and the other fifteen used qualitative methodology (71.4% N = 15). Two studies were mixed (5.3%; N = 2). Some evaluation tools were used in the studies, including questionnaires (15.8%; N = 6), observation (15.8%; N = 6), interviews (29%; N = 11), focus groups (10.5%; N = 4), and documentary analysis (26.3%, N = 10).

In 47.1% (N = 16) the sample was made up of teachers, management teams, or social professionals, in 32.4% (N = 11) the sample was made up of students, in 14.7% (N = 5) of family members, and in 2.9% (N = 1) there were programs designed for the intervention of children with cultural diversity.

All the information can be seen in Table 2, which also lists the authorship, year of publication, purpose, sample, country, methodology with the evaluation tool, and main results.

**Table 2.** Summary of the main characteristics and results of the selected studies.

| Citation | Purpose | Design | Sample | Tools | Primary Results | Country |
|---|---|---|---|---|---|---|
| Gómez Chaparro and Sepúlveda Sanhueza (2022) [50] | To analyze the challenges of school leaders when including migrant students in a Chilean school. To identify school leadership practice that favor the inclusion of migrant students. To understand school leaders' perceptions about the inclusion of migrant students and their strategies to promote it. To identify barriers and challenges in the process of inclusion of migrant students and how school leaders can address them. | Qualitative methodology | A total of three members of the management team (director, the person in charge of school cohabitation, and the social worker); three teachers with migrant students and three inspectors, three teachers of the school integration program, and six parents of foreign origin. | Semi-structured interviews and a focused discussion group. | Three strategies were identified for school leaders to foster the inclusion of migrant students in a Chilean school and establish an inclusive school climate. They provide emotional and social support and adapt pedagogical practice to meet the needs of migrant students. | Chile |
| León et al. (2018) [51] | To know the management tasks that promote inclusion. | Quantitative methodology | A total of 243 families and 154 teaching teams from 17 public and subsidized primary and secondary schools. | Questionnaire: leading inclusive education (LEI-Q). | The management teams of schools in socioeconomically disadvantaged areas carried out fewer inclusive actions than those located in more favored areas. | Spain |
| Jiménez-Vargas (2022) [52] | Analyze how schools manage the tensions between an inclusive model and an advanced neoliberal model, aiming to identify effective strategies to promote inclusion and educational equity. | Qualitative methodology | A total of 4 schools, 23 interviews with education professionals, 5 focus groups of foreign students and families, 32 classroom observations, 7 documents, and 4 shadowing sessions. | Participant observation, documentary analysis, and interviews. | Practice of adaptive resistance was identified which generated three turns that strained the management of the model: affective, collaborative, and localist. The role of teachers in educational inclusion and the importance of diversity in school culture were highlighted. | Chile |
| Borrero López and Blázquez Entonado (2018) [53] | To examine the pedagogical model used in intercultural education by teachers. To identify teacher training needs. To provide recommendations for the development of the critical intercultural approach in early childhood, primary, and secondary schools. | Mixed methodology | A total of 137 kindergarten, primary, and secondary school teachers. A total of 22 programs of attention to immigrant students in Extremadura. | Questionnaire, interviews, focused discussion groups, and document analysis. | Limited teacher perception of multiculturalism. Concern and interest in minority groups. It proposed a new model of intercultural schools based on an interactive model. Create networks between schools, establish learning communities, adapt teacher training to appropriate pedagogical models, achieve committed and involved administration and management teams to provide autonomy to schools, and ensure the quality and equity of the educational response. | Spain |

**Table 2.** *Cont.*

| Citation | Purpose | Design | Sample | Tools | Primary Results | Country |
|---|---|---|---|---|---|---|
| Bugno (2018) [54] | To determine teachers' beliefs about cultural diversity and how they influence planning and teaching practice. | Qualitative methodology | A total of 45 elementary school teachers. | Interview | There is a discrepancy between the theories and normative documents of intercultural education and the beliefs and practice of teachers. There are biases in the teachers' statements which led to a moralizing discourse and the repetition of practice without reflecting on their appropriateness to the specific context. | Italy |
| Garreta-Bochaca et al. (2022) [55] | To analyze the degree of recognition of cultural diversity in the school's documentation, the actions carried out in relation to cultural diversity (in the classroom, in the menus, in the culture and language of origin classes), and the practical development of intercultural educational practice. | Quantitative methodology | A total of 1730 representatives of the management teams of primary education schools. | Questionnaire | The results indicate that there is still a long way to go in intercultural education. Schools with a lower presence of students of foreign origin tend to perceive this issue as less of a priority and act less in this direction compared with other schools. The misconception persists that it should only be addressed in culturally diverse schools. | Spain |
| Antolinez Domínguez and Jorge Barbuzano (2022) [56] | To analyze the models of cultural diversity management through two case studies carried out in Spain and Mexico during the rise of intercultural education. | Qualitative methodology | Two schools, one in Andalusia (Spain) and the other in Oaxaca (Mexico). Key players in the centers: 55 interviews in Spain and 47 in Mexico. | Case studies: school and community ethnographies (observations, semi-structured interviews) | It highlights the importance of identifying the aspects that can function as analytical facets for intercultural education from the paradigm of diversity. The paradigms of inequality and difference are those that underpin educational policies based on culturalism, which has a double logic. First, diversity is reduced to cultural criteria, which can confuse diversity with inequality. Second, cultural homogenization occurs within groups, leading to a distinction between "them" and "us". | Spain and Mexico |
| Dežan and Sedmak (2023) [57] | To analyze the influence of the school environment on the well-being of adolescent migrants, focusing on the pedagogical practice and interpersonal relationships established between them, their peers, and teachers. How these factors may affect their academic, social, and emotional adaptation and development in the school environment. | Qualitative methodology | A total of 700 immigrant adolescents from 46 schools in six countries: Austria, Denmark, Slovenia, Spain, Poland, and the United Kingdom. | MiCREATE project questionnaire | Adolescent migrants feel safe and like school, although they are more satisfied with relationships with teachers than with their peers. Differences in school well-being vary according to the country's migration experience. Fostering intercultural education and satisfying interpersonal relationships are essential for adolescent migrants' school well-being and successful integration. | Austria, Denmark, Poland, Slovenia, Spain, Poland, and the United Kingdom. |

**Table 2.** *Cont.*

| Citation | Purpose | Design | Sample | Tools | Primary Results | Country |
|---|---|---|---|---|---|---|
| Espinosa and Pons (2020) [58] | To analyze the affective facet present in school experiences aiming to understand how emotions, feelings, and affective relationships influence the way students and other social actors perceive and experience education in the region. | Qualitative methodology | Five social agents (co-investigators) | Accounts of school life; interviews, photobiography; visual ethnography; and collective observations. | The affective dimension is increasingly important in educational research. It is necessary to understand and intervene in education at the local level, recognizing intercultural educational regions and their characteristics. | Mexico |
| Ceballos Vacas and Trujillo-González (2021) [59] | To learn about the emotional difficulties of migrant students and the support received through the perception of their educational community and the analysis of the school's documentation. | Qualitative methodology | Four teachers One social worker One social educator Four students (three immigrants and one Canary Islander) Two immigrant mothers | Interviews and document analysis of the educational project of the center (PEC) and the Annual general programming (PGA). | The study shows that the school fosters an effective intercultural culture that celebrates diversity and combats discrimination. There are emotional difficulties for migrant students (migration experience, gender role conflicts, and belonging to segregated groups). Specific teacher training in intercultural and emotional competencies is required. Improve the relationship between the school and migrant families with a more systematic and flexible intervention to achieve diversity-sensitive school participation. | Spain |
| Micó-Cebrián et al. (2019) [60] | To analyze the relationship between personal and school variables (empathy, self-concept, and perceived teacher helpfulness), intercultural sensitivity, and life satisfaction in elementary school students. | Quantitative methodology | A total of 473 primary school students (native and immigrant, between 10 and 13 years of age) | Five scales: Micó-Cebrían and Cava scale (2014); Diener, Emmons, Larsen, and Griffin scale (1985); Jolliffe and Farrington scale (2006); 5-AF5 scale by García and Musitu (1999); and Moos and Trickett scale (1973). | Presence of some significant differences between native and immigrant students in relation to intercultural sensitivity, life satisfaction, empathy, self-concept, and perception of help. Needs were detected in relation to teacher training in interculturality and the development of intervention programs to improve communication and emotional self-concept and the involvement of families and social entities to foster respect and trust among students. | Spain |

**Table 2.** *Cont.*

| Citation | Purpose | Design | Sample | Tools | Primary Results | Country |
|---|---|---|---|---|---|---|
| Sales Ciges et al. (2018) [61] | To identify the resources and strategies used by schools to promote participation and linkage with the environment, and to know the perception of teachers and families about their participation in schools in the Valencian Community, the Basque Country, and the Region of Murcia. | Quantitative methodology | A total of 242 centers (927 teachers and 3845 families) | Resource inventory questionnaire. Family and teacher questionnaire on participation. | The low level of school adaptation of students is related to their family and social environment and the educational attitude of their parents. Cultural, linguistic, and ethnic diversity negatively affects the development of inclusive attitudes and requires training in coexistence to resolve conflicts in the classroom. Lack of coherence between the culture and family environment of the students and their school environment, as well as the influence of the family in the formation and development of the students. | Spain |
| Maiztegui-Onate et al. (2019) [62] | Analyze the process of implementation of activities and reflect on the notion of interculturality in education from a broad perspective related to global citizenship and education for development. | Qualitative methodology | Two primary education centers, with few migrants in the Basque Country. A total of 16 teachers. | Focused discussion group (three groups) and teacher's logbooks (10 logbooks). | Three perceptions of interculturality were distinguished: (a) generalist; (b) focused on cultural aspects; and (c) social justice approach. The intercultural logic is based on assumptions such as equality and justice, which cannot be imposed but must be adopted at a personal level. | Spain |
| Molina Díaz and Sales (2021) [63] | To know the process of building the school's educational community and the factors involved in this process. | Qualitative methodology | Interview of the principal, two teachers, two parents, two students, and one member of the city council. Focused discussion group: four parents, four teachers, and three students. | Interviews, focused discussion group, and participant observation). | The results remark the importance of highlighting the collective identity of the community at the beginning of the construction process and how affiliation to a group is not enough for people to identify with it. The complexity of the collective identity-building process in the community is highlighted. The importance of searching for identity symbols and motivating community members on an ongoing basis is pointed out. Dichotomous discourse that influences the evolution of collective identity. | Spain |

**Table 2.** *Cont.*

| Citation | Purpose | Design | Sample | Tools | Primary Results | Country |
|---|---|---|---|---|---|---|
| Gómez-Hurtado et al. (2021) [64] | To explore good management practice in cultural diversity and attention to immigrant students from a passive leadership perspective. | Qualitative methodology | Six early childhood and primary education centers. Two in Spain and four in Chile (directors and management teams) and with informal leaders (support teachers and non-teaching professionals). | Interviewing, participant observation, and document analysis. | The main results show the importance of leaders in promoting an inclusive collaborative culture in classroom practices focused on the knowledge and cultural capital of foreign students. Management should focus on the development of organizational and didactic strategies based on the recognition and participation of the educational community, its commitment to social justice, collaborative management of diversity, and a shared concept of educational inclusion. | Spain Chile |
| Torrelles Montanuy (2022) [65] | To analyze the predominant approaches and actions in the practical development of intercultural education in primary education in Spanish schools considering the ownership and the percentage of students of foreign origin in each of them. | Mixed methodology | A total of 1730 primary school students; 19 interviews with teachers from five schools. | Questionnaire and interview. | The results show that the presence of foreign students is the most influential variable in the implementation of intercultural education in schools. Although there is a discursive recognition of the value of intercultural education, no actions with an intercultural perspective are identified in its practical application in most schools. Democratic participation in the community is essential and is related to co-responsibility and shared sovereignty. The relationship between democratic participation, collaborative culture, and distributed leadership in the inclusive educational community is highlighted, and the importance of learning to participate is emphasized. | Spain |
| Gromova et al. (2021) [66] | To identify the educational practice used by elementary school teachers to provide linguistic and academic support to immigrant children and to promote a welcoming classroom environment that favors their psychological adjustment. | Qualitative methodology | A total of 20 elementary school teachers who have experience with immigrant students. | Interview | Teachers do not use special methods for teaching Russian as a foreign language and recognize the need for training in teaching methods for immigrant students. There is a need for teacher training programs that include teaching multilingual and culturally diverse students in the classroom. | Russia |

**Table 2.** *Cont.*

| Citation | Purpose | Design | Sample | Tools | Primary Results | Country |
|---|---|---|---|---|---|---|
| Traver Martí et al. (2019) [67] | To describe the development of the learning and service project (ApS) as a curricular practice linked to the territory. To analyze the strategies of curriculum negotiation and student participation in this educational practice. | Qualitative methodology | Students in sixth grade of primary education, two teachers, the families involved in the ApS, the school's management team, and the education councilor of the city council. | Participant observation, interviews, focused discussion groups, audiovisual records, documentary analysis, and field diary. | The ApS curricular practice carried out among all sectors of the educational community, through cooperative dynamics and participatory social diagnoses such as social mapping, cooperative learning, and class assemblies, has allowed students to reach their maximum learning potential and achieve true democratic participation. The educational experience has enabled all participants to become critically and actively aware of their reality and has led to a transformative and emancipatory education. The leadership exercised by the teachers is within the parameters of distributed leadership, central to the experiences of democratic schools and social justice models. | Spain |
| Guzmán Vargas et al. (2020) [68] | To identify a pedagogical strategy that would allow the interaction of the Wounaan and Mestizo cultures present in the classrooms of a school in Bogota. | Qualitative methodology | A total of 27 students (15 belong to the Wounaan indigenous community) of the Compartir Recuerdo IED school (Bogota). | Action research (field diaries and interviews). | The project "Sharing our worlds (Project-Based Learning—ABP): Construction of a mural through collaborative work, sharing experiences, languages, cultural diversity and ways of life with empathy, solidarity and respect for diversity" is presented. Intercultural processes have been achieved in students, parents, and teachers, which highlights the importance of pedagogical strategies that promote the formation of individuals who transform formative dynamics based on cultural diversity. It is necessary to reconstruct the meaning of inclusive education in order to take into account cultural exchanges in the classroom and allow the general population to achieve human dignity and quality of life. | Colombia |

**Table 2.** *Cont.*

| Citation | Purpose | Design | Sample | Tools | Primary Results | Country |
|---|---|---|---|---|---|---|
| Gómez Hernández et al. (2022) [69] | To describe and understand how intervention in cooperative learning and the use of cell phones in kindergarten and primary school classrooms favors school coexistence in a context of socioeconomic vulnerability and cultural diversity. | Qualitative methodology | A total of 52 students from two kindergarten classrooms (five years old) and their respective tutors, and 55 students from two primary school classrooms (six to seven years old) and their respective tutors. | Participant case study, semi-structured interviews, and focused discussion group. | The results point to a reduction in disruptive behaviors, academic disaffection, and exclusion, which is mainly attributed to the combination of some cooperative elements that are favored by the cell phone (positive interdependence, sense of belonging, heterogeneous grouping, individual responsibility, promoting interaction, group identity and cooperative skills). The educational use of cell phones with a cooperative methodology improves school coexistence. | Brazil |
| Mela-Contreras et al. (2022) [70] | To encourage dialogue and critical reflection among teachers and students on the respect and appreciation of cultural and ethnic diversity through cinematography. | Qualitative methodology | Five public schools, 11 teachers, and 145 seventh grade students in visual arts and history, geography, and social sciences. | Documentary and narrative analysis of film language. | The Cineduk program generated in the students the acquisition of adequate competencies to interpret and (re)signify the diversity of the world (ethnic, sexual, and gender). It advocates cinematographic praxis to enrich teaching practice and learning through the curricular and cultural content of students. The development of cinematography as a didactic strategy promotes non-discriminatory and dialogic teaching. | Chile |

Based on the dimensions established for the analysis of the documents, Figure 2 shows the distribution of studies by dimensions/fields. As can be seen, most of the studies focused on organizational aspects (management and leadership) and the pedagogical approach, while the aspects centered on the processes of building a participatory/collaborative school climate were the least frequent studies.

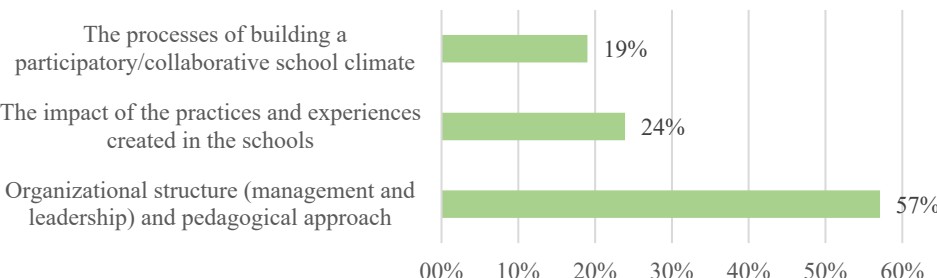

**Figure 2.** Distribution of studies by dimension.

## 5. Discussion

To facilitate the reading and discussion of the results, we display the analysis arranged according to the dimensions that made it possible to identify the main lines of work developed by the schools in the framework of intercultural education.

(1) Regarding the processes of organization and management of the educational response and pedagogical approaches, it should be noted that several studies highlight the importance of school leadership and the adaptation of pedagogical practice in promoting the inclusion of migrant students and the implementation of an alternative intercultural education model focused on the needs of migrant groups [50,64]. It has been observed that management teams in socioeconomically disadvantaged areas have more difficulty carrying out inclusive actions [51], suggesting the need for support and improvement plans that are more ecological and sensitive to the nature of the community to which it intends to respond.

Studies such as [52] highlight the role of teachers in educational inclusion and the importance of training models focused on fundamental aspects for the improvement of attitudes and respect for cultural diversity. The coexistence of different models for the promotion of respect for cultural diversity generated resistance practice that strained the management of a model truly adequate to the needs of the context. In this sense, the adaptation of pedagogical practice is essential to meet the needs of migrant students [50]. The presence of foreign students is a key factor in the implementation of intercultural education in schools [65], but it is also essential to overcome the misnamed "folklorization" that has led to the identification of different cultures by reference to their traditions, gastronomy, dances, etc. [65].

The construction of a collective identity in the educational community is a dynamic and complex process in which all members of the community must participate [63]. The existence of a dichotomous discourse in intercultural education can lead to cultural homogenization within groups and a distinction between "them" and "us" [56]. As [54] points out, there is a discrepancy between the theories and normative documents of intercultural education and the beliefs and practice of teachers, which can generate contradictions and make it difficult to effectively implement intercultural education. Taken together, these findings suggest that it is crucial for schools to adopt an inclusive pedagogical approach tailored to the needs of migrant students and to promote cultural diversity supported by committed school leadership and adequate teacher training in intercultural education [50,52,64]. It is essential that the educational administration provides support and specific improvement plans to schools in socioeconomically disadvantaged areas to facilitate the implementation of inclusive actions [51]. Intercultural education should be extended to the entire student body and educational community, fostering the construction of a collective identity that values and celebrates diversity [63,65]. It is necessary to promote teacher training in inter-

cultural competencies and to encourage critical reflection on own beliefs and practice in relation to cultural diversity [56].

(2) Concerning the situation of schools in relation to creating participatory and collaborative communities, with a climate favorable to intercultural education, the selected studies highlight the importance of building a participatory and collaborative school climate to promote the well-being and inclusion of migrant students [57]. Intercultural education and successful interpersonal relationships between teachers and students are fundamental to achieving this goal [58]. Likewise, it is necessary to address the specific difficulties faced by migrant students, such as migratory experience, gender role conflicts, and belonging to segregated groups [59]. Overcoming these challenges requires a caring school culture and teacher training in intercultural and emotional competencies, as well as in the collaborative relationship with migrant families [59].

Promoting intercultural sensitivity among native pupils and paying attention to the academic self-concept and life satisfaction of immigrant students are key elements to improve school coexistence and the educational inclusion of this group [60]. The strategies identified by the different studies analyzed include fundamentally: teacher training in interculturality, the development of intervention programs to improve communication and emotional self-concept, and the participation of families and social entities to promote respect and trust among students [60].

In this sense, the affective dimension emerges as a fundamental aspect of educational research, emphasizing the importance of understanding and intervening in education at the local level by recognizing intercultural educational regions and their particularities [58]. Establishing horizontal relationships in the process can expand the possibilities of understanding the cultural meanings and senses that guide people's actions [58].

(3) Where the practical and experiential reality developed from schools on intercultural education is concerned, the selected studies show the impact of the practice and experiences created in schools to generate intercultural learning. For example, ref. [66] examines the teaching methods used in Russia to teach Russian to immigrant children and highlights the need for training in teaching methods for immigrant students. The findings suggest that teacher training programs are needed that include teaching multilingual and culturally diverse students in today's classrooms [66].

According to [67], educational practice allows students to reach their full learning potential and achieve true democratic participation. These educational experiences promote a transformative and emancipatory education, and the leadership exercised by teachers is recognized as essential to the experiences of democratic schools and social justice models [67].

Where [68] is concerned, there is an emphasis on the importance of pedagogical strategies that promote the development of transforming individuals of formative education dynamics based on cultural diversity. Pedagogical proposals should be based on relational, functional, and critical interculturality to generate spaces of exchange considering the knowledge of each student [68]. As examples of practical strategies, ref. [69] indicates that the educational use of mobile phones with a cooperative methodology is a good example of a useful strategy to improve school cohabitation and reduce disruptive behaviors, academic disaffection, and exclusion [27]. On the other hand, refs. [70,71] are committed to the use of cinematography as a didactic strategy to promote non-discriminatory and dialogic teaching by using film language in audiovisual creations on cultural diversity.

## 6. Limitations and Future Work

One of the main limitations found in this study focuses on the nature, objectives, and scope of the interventions around the working model in intercultural education. The evidence points to the need to consider new organizational models for managing and approaching pedagogical styles focused on the construction of an intercultural school climate. Isolated, sporadic interventions are disconnected from the organizational and ped-

agogical development of the centers without a global and integrating vision of intercultural education within the curricular design of them.

These are highly frequent and ineffective, as they only cover areas of the curriculum and are not integrated into the culture of intercultural inclusion. The reality of the intercultural fact of the centers is hardly visible, few experiences are shared, and the real scope of the actions carried out in the schools is unknown.

The specific and particular initiatives that are being proposed could be redirected toward the construction of innovations at the center or institutional level and could be triggered for more global actions that mobilize the entire educational community. In this way, the deployment of inclusive intercultural education could result in repercussions of greater scope, reach, breadth, and significance.

It is recommended that future studies consider the experiences of centers, the pedagogical principles of which contemplate the educational response to intercultural reality. Field studies could improve the analysis, quality, and scope of the experiences that have been collected in the studies presented in this manuscript. These studies would provide detailed information on how this practice is being applied in schools and how it is perceived by students, teachers, and families. Therefore, the knowledge of this practice could be deepened and provide more precise and effective recommendations for educational leaders who wish to implement strategies to foster a culture of inclusion and diversity in their institutions.

## 7. Conclusions

This study dealt with three key dimensions in the search for inclusive and intercultural education in diverse school environments. Through an updated systematic literature review, management reorganization processes and pedagogical approaches, participatory and collaborative school climate construction, as well as the impact of practice and experiences created in schools to generate intercultural learning were assessed.

The study provides a comprehensive view of the advances and challenges in inclusive and intercultural education in diverse school sceneries. Although significant progress has been made in implementing inclusive practice and building participatory and collaborative school climates, there are still areas that require greater attention and development.

One of these areas is the ongoing training of teachers in intercultural competencies and socio-emotional skills. The inclusion of cultural diversity into the curriculum and the adoption of inclusive pedagogical approaches not only enhances the educational experience of students but also promotes fairness and social justice in the educational system. In addition, it is essential to strengthen the relationship between schools and families to support the process of integration and adaptation of migrant students and their families to the school environment.

It is important to continue researching and evaluating the impact of educational practice and experiences on the generation of intercultural learning. Identifying and promoting effective pedagogical strategies and innovations in teaching and learning can contribute to a more inclusive and fair education. In addition, it is essential to promote the participation and collaboration of all actors involved in education, including students, parents, teachers, administrators, and educational leaders, as well as political authorities and civil society organizations.

In summary, this study provides an updated and comprehensive overview of the situation of inclusive and intercultural education in diverse school environments, linking the results obtained in the three dimensions with the objectives. The findings can help future research and practice in this field, as well as inform educational policies and strategies to address cultural diversity in schools. Through an interdisciplinary and collaborative approach, it is possible to move towards a more inclusive, equity-based, and socially fair educational system that values and celebrates cultural diversity as a stimulating resource for all students.

**Author Contributions:** Conceptualization, D.P.-J. and A.I.G.-H.; methodology, D.P.-J., A.I.G.-H. and M.G.-A.; validation, D.P.-J., A.I.G.-H and M.G.-A.; formal analysis, A.I.G.-H., M.G.-A. and A.G.S.-Á.; investigation, D.P.-J., A.I.G.-H. and M.G.-A.; resources, A.I.G.-H. and A.G.S.-Á.; D.P.-J., A.I.G.-H. and M.G.-A.; writing—original draft preparation, D.P.-J. and A.I.G.-H.; writing—review and editing, A.I.G.-H., M.G.-A. and A.G.S.-Á.; visualization, D.P.-J., A.I.G.-H., M.G.-A. and A.G.S.-Á.; supervision, D.P.-J. and A.I.G.-H.; project administration, A.I.G.-H. and M.G.-A. All authors have read and agreed to the published version of the manuscript.

**Funding:** This research received no external funding.

**Institutional Review Board Statement:** Not applicable.

**Informed Consent Statement:** Not applicable.

**Data Availability Statement:** Information and queries on the data used can be obtained from this article.

**Conflicts of Interest:** The authors declare no conflict of interest.

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
