# Peer review of "Reality and Future of Interculturality in Today’s Schools"

_education, doi:10.3390/educsci13050525_

Round 1

Reviewer 1 Report

Both the abstract and the keywords are as expected. Thus, the summary has an overview of the paper with background, purpose of the study, methods, synthesis of the main discoveries, as well as the conclusions that indicate the concern for designing intercultural educational contexts as a challenge for schools and their teachers. . An objective representation of the article is perceived.

The introduction opens in a broad context of narration and understanding, highlighting the importance of the issue of interculturality in the formal school, with the difficulties that the conceptualization of the term interculturality has. This has led to background reviews on the subject with adequate and sufficient citations. There are references in relation to education and close to globalization, highlighting the multiethnic and multilingual nature of countries and diverse cultures. Interculturality is addressed in various contexts, as well as the evolution that has been taking place in the concept of interculturality, until reaching the inclusive approach to intercultural education at school and, in all cases, with appropriate references. The importance of promoting intercultural education is highlighted. In the same way, the general objective and the specific ones that go through the review of the processes and the analysis of these, carried out by the educational communities, are specified.

The manuscript specifies the materials and the study method, where the review is carried out with the protocol of the updated PRISMA declaration, which addresses the dimensions of the construction processes of the participatory school climate, the organizational and pedagogical structure of inclusive schools and the impact of the practices and experiences created in these inclusive intercultural schools, with searches of different terms associated with interculturality in the main databases in English and Spanish.

The inclusion and exclusion criteria are indicated in Table 1, which favored an adequate selection of documents for the study. The considerable number of results obtained (3755) and the small final number of documents selected (21) after applying the indicated criteria are striking.

Regarding the specific procedure followed with the documents selected for the review, the methodology followed in each case, the type of evaluation instruments, as well as other complementary details are indicated.

The analysis reflects that the majority of the studies focused on organization, management and pedagogical style issues, being infrequent to address the construction of the school climate in the intercultural model.

The discussion is the section of the study that offers more development and contribution of results interpreted and contrasted in perspective with other previous studies. Thus, the three main elements are perceived: a) the organization and management processes of the educational response and the pedagogical approaches; b) schools with participatory and collaborative communities in relation to the construction of a climate that favors intercultural education; and c) the reality that exists in schools with intercultural education, which require specific teacher training programs with democratic educational experiences with practical strategies. In the three sections of the discussion, references are shown to related studies with which a discussion is established.

The conclusions focus on the three known dimensions for inclusive and intercultural education in diverse school settings. The progress achieved by implementing inclusive practices and building a climate of participatory and collaborative coexistence is highlighted. The need for continuous training of teachers in this area of competence, the inclusion of curricular cultural diversity, emphasizing equity and social justice, is verified. The need to investigate, evaluate and make visible intercultural educational practices and experiences is underlined.

I miss the specification of some limitations perceived in the study and the specification of other complementary research in the future.

The findings that are offered can be indicative for organizations, especially schools, that address cultural diversity, from an interdisciplinary and collaborative competence approach, for which I congratulate the authors who have successfully proposed and developed this manuscript.

Author Response

Thank you for your positive review of our manuscript.

Regarding the limitations of the study, we have added a section on limitations and future lines of work and research.

One of the main limitations found in this study focuses on the nature, objectives, and scope of the interventions around the working model in intercultural education. The evidence points to the need to think of new organizational models for the management and approach of pedagogical styles focused on the construction of an intercultural school climate. Isolated, sporadic interventions that are disconnected from the organizational and pedagogical development of the centers, without a global and integrating vision of intercultural education within the curricular design of the centers.

 These are highly frequent and ineffective, as they only cover areas of the curriculum and are not integrated into the culture of intercultural inclusion. The reality of the intercultural fact of the centers is little visible, few experiences are communicated and the true scope of the actions carried out in the schools is unknown.

The specific and particular initiatives that are being proposed could be redirected toward the construction of innovations at the center or institutional level and could be triggers for more global actions that mobilize the entire educational community. In this way, the deployment of inclusive intercultural education could result in repercussions of greater scope, reach, breadth, and significance.

It is recommended that future studies take into account the experiences of centers whose pedagogical principles contemplate the educational response to intercultural reality. Field studies could improve the analysis, quality, and scope of the experiences that have been collected in the studies presented in this manuscript.

English has been edited by an expert

Reviewer 2 Report

In spite of compreshensive literature review, this study doe not finally include  or analyze documents from Asia and North America. In other word, this study seems to focus on documents from South America or partially Europe area. Thus, this study needs to mention why this geographical areas are omitted. 

None 

Author Response

Thank you for your appreciation

A search for studies was conducted internationally, this included countries all over the world in the first search. The selected manuscripts (more than 200 articles and more than 1,000 abstracts) came from Chile, Ecuador, Colombia, Spain, and Italy, some from Argentina, Italy, and Germany, and Russia, New Zealand, Australia, Sweden, and Russia. The analysis of the manuscripts focused on the objective of this study, left out of the selection many countries from South America or Europe mainly because they did not focus on the processes of building participatory school climate, the organizational and pedagogical structure of inclusive schools, and the impact of practices and experiences created in these inclusive intercultural schools.

The application of the exclusion criteria limited the presence of studies from South America or Europe because they did not focus on the processes of building a participatory school climate, the organizational and pedagogical structure of inclusive schools, and the impact of the practices and experiences created in these inclusive intercultural schools.

To clarify this issue, a section on the limitations of the study has been added to explain this issue.

English has been edited by an expert

Round 2

Reviewer 2 Report

None

Author Response

Thank you very much for your input and comments. We have proceeded to improve the title and have introduced the suggested changes. The manuscript has been reviewed and edited by an English specialist.
If you need the edited English version, we will provide it to you as proof of the improvement of the manuscript.
Kind regards
